
# Evolution of fractality in magnetized plasmas

Víctor Muñoz[1], Macarena Domínguez[2], Juan Alejandro Valdivia[1,3], Simon Good[4], Giuseppina Nigro[5], and Vincenzo Carbone[5]

[1]Departamento de Física, Facultad de Ciencias, Universidad de Chile, Chile
[2]Departamento de Física, Facultad de Ciencias Físicas y Matemáticas, Universidad de Chile, Chile
[3]Centro para la Nanociencia y la Nanotecnología, CEDENNA, Chile
[4]The Blackett Laboratory, Imperial College London, UK
[5]Dipartimento di Fisica, Università della Calabria, Italy

*Correspondence to:* Víctor Muñoz (vmunoz@fisica.ciencias.uchile.cl)

**Abstract.** We studied the temporal evolution of fractality for geomagnetic activity, by calculating fractal dimensions from Dst data and from an MHD shell model for a turbulent magnetized plasma, which may be a useful model to study geomagnetic activity under solar wind forcing. We show that the shell model is able to reproduce the relationship between the fractal dimension and the occurrence of dissipative events, but only in a certain region of viscosity and resistivity values. We also

present preliminary results of the application of these ideas to the study of the magnetic field time series in the solar wind during magnetic clouds. Results suggest that the fractal dimension is able to characterize the complexity of the magnetic cloud structure.

*Copyright statement.* TEXT

## 1 Introduction

The nontrivial interaction between the Sun's and the Earth's magnetosphere, coupled by the solar wind, leads to a rich variety of phenomena which has attracted interest to the study of space plasmas for decades, and more recently to the possibility of forecasting of space weather, an issue of large relevance in our increasing technology-dependent society.

Various models and techniques have been developed to study the plasma behavior in the Sun-Earth system. Of these, the study of complexity has been of great interest, as they are capable of providing new insights and reveal possible universalities

on issues as diverse as geomagnetic activity, turbulence in laboratory plasmas, physics of the solar wind, among others. (Dendy et al., 2007; Klimas et al., 2000; Takalo et al., 1999; Chang and Wu, 2008; Valdivia et al., 1988). In particular, these studies have shown that systems such as the magnetosphere (Chang, 1999; Valdivia et al., 2005, 2003, 2006, 2013), the solar wind (Macek, 2010), the solar photosphere, and solar corona (Berger and Asgari-Targhi, 2009; Dimitropoulou et al., 2009), are in a self-organized critical state, and exhibit complex features such as fractality and multifractality. Some authors have discussed the

relationship between the fractal dimension, as a measure of complexity, and physical processes in magnetized plasmas in the Sun-Earth system, including the possibility of forecasting geomagnetic activity (Aschwanden and Aschwanden, 2008; Uritsky





et al., 2006; Georgoulis, 2012; McAteer et al., 2005, 2010; Dimitropoulou et al., 2009; Conlon et al., 2008; Chapman et al., 2008; Kiyani et al., 2007).

In our work we use the box-counting fractal dimension (Addison, 1997), because of its simplicity and its intuitive meaning. Certainly, a single fractal dimension cannot provide all information on complexity for arbitrary systems, in particular if they also exhibit multifractal behavior as well, as expected in the magnetospheric system (Chang, 1999), models of turbulence (Kadanoff et al., 1995; Pisarenko et al., 1993), and the solar wind (Chapman et al., 2008); but it is interesting to note that it does describe some relevant features of these time series' complexity, as it has been successfully used in previous works relevant to the Sun-Earth system (Osella et al., 1997; Kozelov, 2003; Gallagher et al., 1998; Georgoulis, 2012; Lawrence et al., 1993; Cadavid et al., 1994; McAteer et al., 2005). Furthermore, the box-counting dimension is a fast approach to systematically study our systems of interest, and a first step to detect universal features worth of further study.

It is also worth noting that the fractal dimension we calculate is based on a scatter diagram (see *e.g.* (Witte and Witte, 2009)), whereas previous studies have been done with other methods or data (Kozelov, 2003; Uritsky et al., 2006; Balasis et al., 2006; Dias and Papa, 2010).

These ideas were implemented by us in Ref. (Domínguez et al., 2014) to study the *Dst* time series and solar magnetograms, and the possible correlation between solar and geomagnetic activities as evidenced by the box-counting fractal dimension. Individual events, complete years of high geomagnetic activity, and the full 23rd solar cycle were studied with this technique, successfully finding that the fractal dimension, and more specifically its evolution, has —despite its simplicity— relevant information on the complex behavior of these systems and their eventual correlation.

Results above were robust, in the sense that they were observed across a wide range of time scales, which suggests that any model describing the dynamics of geomagnetic activity should reproduce a similar fractal behavior. This is our motivation to study a shell model for MHD turbulence within this framework.

Evidence of turbulence in the Earth's magnetosphere has been found by various spacecraft observations (Nykyri et al., 2006; Sundkvist et al., 2005; Zimbardo et al., 2008), and several authors have studied magnetospheric MHD turbulence (see, *e.g.*, Borovsky (2004); Hwang et al. (2011); El-Alaoui et al. (2012)). One interesting approach has been the proposal of analytical models depending on few degrees of freedom, which nevertheless retain relevant statistical properties of the magnetospheric behavior, such as the power-law distribution and multifractal features of dissipative events (Chapman et al., 1998; Valdivia et al., 2006).

Shell models constitute an intermediate level between such models and first principles approaches. They are low dimensional models, based on a system of coupled equations mimicking the spectral Navier-Stokes equation, and have been used to describe turbulence in magnetized fluids, describing the main statistical properties of magnetohydrodynamic (MHD) turbulence (Chapman et al., 2008), without the computational cost of performing high Reynolds numbers simulations directly from the fully nonlinear fluid equations (Ditlevsen, 2011).

Dissipative events in shell models have been shown to follow the same power-law statistics of observed events in turbulent magnetized plasmas, as found in Refs. Boffetta et al. (1999); Lepreti et al. (2004); Carbone et al. (2002), where dissipative


events in the model were taken to represent solar flares. In fact, these works suggest that flares and geomagnetic activity should be the result of dissipation bursts within a turbulent environment (Lepreti et al., 2004)

In a previous work (Domínguez et al., 2017), we have applied the box-counting fractal dimension to study the complexity in an MHD shell model, analyzing the correlation between it and the energy dissipation rate, showing that, for certain values

of the viscosity and the magnetic diffusivity, the fractal dimension exhibits correlation with the occurrence of bursts, similar to what had been found with geomagnetic data (Domínguez et al., 2014). This suggests that shell models do not only reproduce the power-law statistics of dissipative events in turbulent plasmas, but also some features of its fractal behavior.

In this manuscript we review our results in this field, where the fractal dimension is calculated in order to measure complexity in magnetic field times series. The method is used to characterize the occurrence of events such as geomagnetic storms by

means of analyzing the *Dst* time series in various time scales (described in Secs. 2–8, and discussed previously in more details in Domínguez et al. (2014)), and the occurrence of dissipative events in an MHD shell model simulation (Secs. 6–7, see Domínguez et al. (2017) for more details). We also present preliminary results dealing with spacecraft data for the solar wind, related to the appearance of magnetic clouds (Muñoz et al., 2016) (Sec. 8).

## 2 Fractal dimension

We are interested to estimate the fractal dimension to various time series for magnetic data. We now explain the method, using as an example the hourly *Dst* time series (World Data Center for Geomagnetism, Kyoto).

There are various ways to define a fractal dimension for a time series (Addison, 1997; Theiler, 1990). Although there is no simple way to relate different definitions, in general it can be said that they are noninteger numbers measuring the complexity of a data set. In this work, we estimate the fractal dimension using the box-counting method (Addison, 1997) in the way we

now describe. First, we construct a scatter diagram for each *Dst* time series. If $Dst^i$ is the $i$-th *Dst* datum in the series and $N$ is the total number of data, the scatter diagram is a plot of $Dst^{i+1}$ versus $Dst^i$, for $1 \le i \le N - 1$, as shown in Fig. 1.

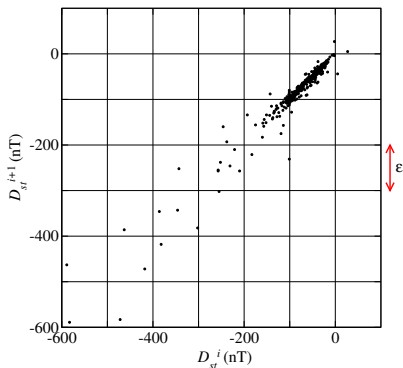

**Figure 1.** Scatter diagram for the hourly *Dst* time series corresponding to the first storm state (6 to 20 March) 1989. (More details in Sec. 3.) The size of the square box is $\epsilon$.





Then, the scatter diagram is divided into square cells of a certain size $\epsilon$, and we count the number $N(\epsilon)$ of cells which contain a point belonging to the set. Finding the range of values of $\epsilon$ where $\log(N(\epsilon))$ scales linearly with $\log \epsilon$, the scatter diagram box-counting dimension $D$ is then defined by the slope in this linear regime, that is,

$$N(\epsilon) \propto \epsilon^{-D} , \tag{1}$$

We estimate the error in $D$ through the least squares fit for the slope.

Further details and discussion on the method can be found in Ref. Domínguez et al. (2014).

It is clear that, in order to calculate $D$, a certain time frame of the dataset must be chosen. Given the time windows chosen for *Dst*

The method as stated above was applied to the *Dst* time series where, given the width of the data windows used (the criterion
is discussed in Sec. 3) and the time resolution of the data (one point per hour), it only made sense to build the scatter plot with consecutive data points.

However, when resolution is larger, as is the case with simulation and solar wind data, it is possible to consider different time delays. Thus, the scatter plot can be built by plotting the $i$-th data in the set, versus de $(i+j)$-th data, with $j \geq 1$ in general, and then the fractal dimension calculated depends on $j$, $D_j$. This was the approach in Refs. Domínguez et al. (2017) and Muñoz
et al. (2016), and presented here in Secs. 6 and 8.

## 3   *Dst* time series: Storm and quiet states

We first apply this technique to quiet and active periods with magnetic storms in order to investigate the relationship between the intensity of the *Dst* index and its fractal dimension, a relationship which has also been suggested by other studies of the complexity of the *Dst* series. (Balasis et al., 2009; Papa and Sosman, 2008)
Following Ref. Domínguez et al. (2014), we identify "storm states" and "quiet states" by locating peaks in the *Dst* series where $D_{st} < -220$ nT, and then a "storm state" is defined by a window starting one week before the minimum value of the peak, and ending one week after it. This is done considering the typical time scale of a geomagnetic storm (Tsurutani and Gonzalez, 1994; Gonzalez et al., 1994). Then, the "quiet state" corresponds to the period of time between two "storm states". Figure 2 illustrates this by showing the four peaks detected in 1989 and the corresponding windows.
For future identification, we label each state in a year with consecutive integer numbers, starting from 1. For instance, in Fig. 2, the year starts with a quiet state, then that will be state "1"; the following state will be a storm, and it will be state "2". Thus, all future quiet states within the year will be labeled with consecutive odd numbers, whereas storm states will be labeled with consecutive even numbers.

The box-counting dimension for each storm and quiet state, calculated as described in Sec. 2, is shown in Fig. 3. Red circles
indicate storm states. Error bars in $D$ are given by the error of its least squares linear fit.

Similar plots for 5 years of high geomagnetic activity were obtained (Domínguez et al., 2014). In general, it is found that storm states have smaller fractal dimension than the surrounding quiet states, although there does not seem to be a clear





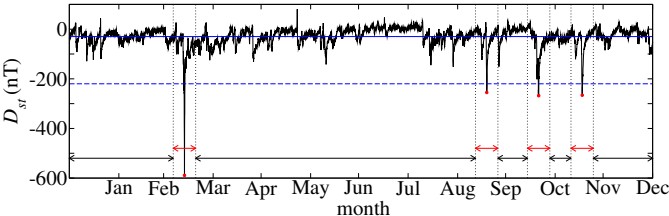

**Figure 2.** *Dst* time series for 1989, identifying the storm and quiet states as explained in Sec. 3. The solid horizontal line shows the average value, and the dashed horizontal line the threshold value used to identify a geomagnetic storm. Red dots show the minimum *Dst* value used to identify a "storm state". Red and black arrows show windows corresponding to storm and quiet states, respectively.

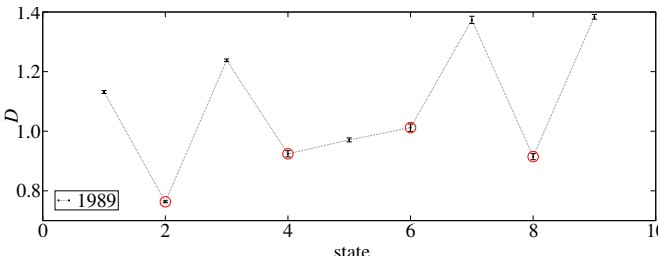

**Figure 3.** Box-counting dimension $D$ for storm and quiet states for year 1989. The abscisa represents the labeling of the states as explained in Ref. 3. Red circles indicate storm states.

correlation on the value of *Dst* itself, and the fractal dimension, as shown in Fig. 4 for all states, for all years studied in Domínguez et al. (2014). No obvious correlation is found if individual years are considered either. Thus, our statement on the decrease of the fractal dimension is an argument on its variation, rather than on its actual value.

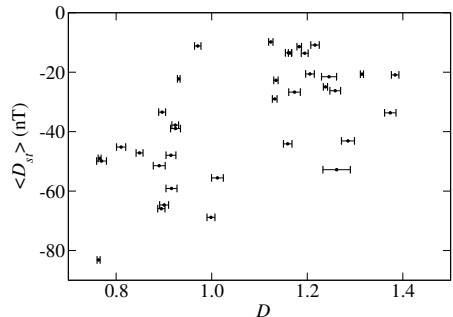

**Figure 4.** Mean value of *Dst* for each state as function of the box-counting dimension $D$ with respective error bars (calculated as in Fig. 3), for five years of high geomagnetic activity: 1960, 1989, 2000, 2001, and 2003.



## 4   *Dst* time series: Variable width windows around a storm

If the qualitative connection between fractal dimension and existence of a storm observed in Sec. 3 is robust, then widening the window around a storm should increase its fractal dimension, as more "quiet" data are taken into account.

To this end, we take windows starting/ending $n$ weeks before/after the peak, with $n = 1, \ldots, 6$. We illustrate this in Fig. 5,
5    where the windows considered around the 13 March 1989 storm are shown.

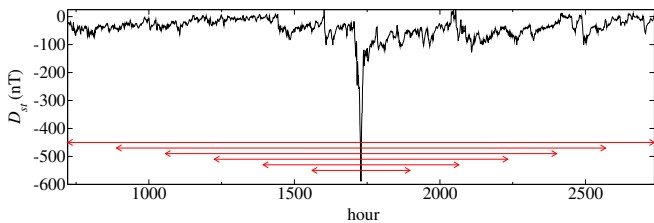

**Figure 5.** Variable size windows around the 13 March 1989 storm (peak at abscissa 1729). The plot shows the *Dst* index as a function of time, measured in hours since the beginning of the year.

Figure 6(a) shows the results for four particular storms: 1 April 1960, 13 March 1989, 6 April 2000, and 30 March 2001, with minimum intensities of $-327$ nT, $-589$ nT, $-288$ nT, and $-387$ nT, respectively. These storms have been chosen because they are isolated, so that windows can be enlarged (up to four weeks on each side) without including new "storm states" (Domínguez et al., 2014).

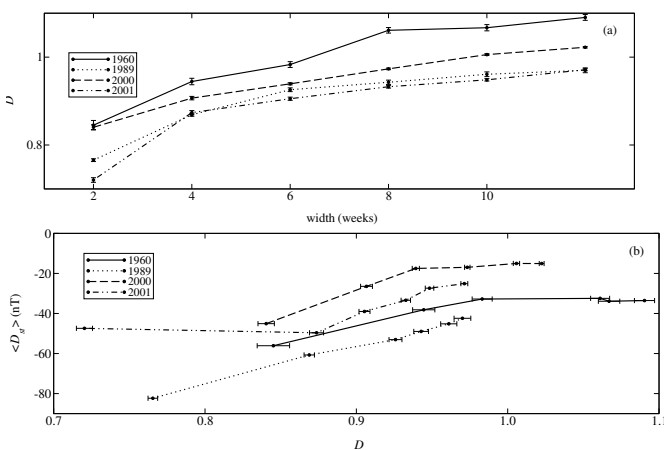

**Figure 6.** (a) Box-counting dimension $D$ for a storm state with respective error bars, as a function of the width of the window around it. (b) Mean value of *Dst* for each variable width window around the same storms in (a), as function of the box-counting dimension with respective error bars.

10    Consistent with the results in Sec. 3, the box-counting dimension increases as we zoom out from the storm, which means that the relevance of the storm itself within the window decreases. This is confirmed by plotting the mean value of *Dst* in



a window as a function of $D$, for the same storms. This is shown in Fig. 6(b). We expect that increasing the window width should increase not only the value of $D$ as noted above, but also the average value of $Dst$ for the same reason, and thus $D$ and $\langle Dst \rangle$ should be positively correlated. This is confirmed in Fig. 6(b). The breaks in the linear behavior for some curves can be explained by the existence of nearby peaks close to the storm studied, as explained in detail in Domínguez et al. (2014).

## 5   *Dst* time series: Moving windows across a storm

We now calculate the fractal dimension for fixed width windows (two weeks), initially placed well before the storm peak, and move it in steps of one week crossing the peak. This will give us a better intuition on the evolution of the fractal dimension in time, in particular during a storm. The initial position of the window is the first day of the year, and it is moved until it reaches the third week after the peak (see Fig. 7).

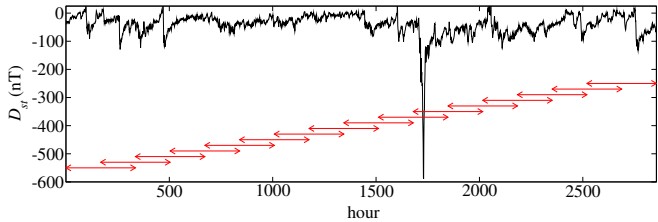

**Figure 7.** Moving windows across a storm (13 March 1989).

Figure 8 shows the results for the fractal dimension for the 13 March 1989 storm, comparing it with the *Dst* index.

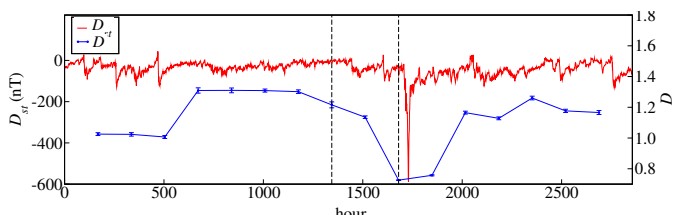

**Figure 8.** Box-counting dimension $D$ (blue, with error bars) and *Dst* index (red) for the 13 March 1989 geomagnetic storm. Vertical lines show windows of data where $D$ decreases before the storm.

For all cases studied (Domínguez et al., 2014), the box-counting dimension of the *Dst* index decreases as the storm approaches. However, it is very interesting to note that we have a noticeable change in the fractal dimension, even before the window contains any point of the geomagnetic storm. This is illustrated in Fig. 8, where two vertical lines indicate the window of *Dst* immediately before the storm. The storm is not included in the window, however the fractal dimension has already started to decrease.

In Domínguez et al. (2014), systematic calculations of cross correlation between *Dst* and $D$ were performed for all storms analyzed, and for the same five complete years studied in that paper (1960, 1989, 2000, 2001, 2003), which have already been



analyzed, but only near geomagnetic storms. Results suggest that the box-counting dimension consistently decreases when the storm approaches, thus suggesting that the box-counting dimension of the *Dst* series, or similar measures of complexity, could be of relevance when forecasting geomagnetic storms.

We also studied the possible correlation between the fractal dimension and measures of solar activity, to investigate whether this simple measure of complexity yields any information about the connection between solar and geomagnetic activities. In particular, we considered the solar flare index (Ataç and Özgüç, 1998; Özgüç et al., 2003) and the coronal index (Rybanský et al., 2001; National Geophysical Data Center (NOAA), Solar Data Services), which are measures of energy released from the Sun.

Results are shown in the left panel of Fig. 9 for the solar flare index, and in the right panel of the same figure for the coronal index, using a moving windows approach over the 13 March 1989 storm, the same storm we have described in the previous sections. Similar analyses were performed for events in 2000 and 2001, as shown in Domínguez et al. (2014). It is found that even for solar flare events of different intensities, periods of large solar flare index are accompanied by a decrease in the fractal dimension $D$ of the *Dst* time series. In the case of the coronal index, results suggest that one or two weeks before the minimum value of $D$, which corresponds to the storm, there is a maximum in the coronal index. However, this is only clearly seen regarding positions of maximum/minimum values. A more detailed correlations analysis, using daily coronal index data does not show any particular signature.

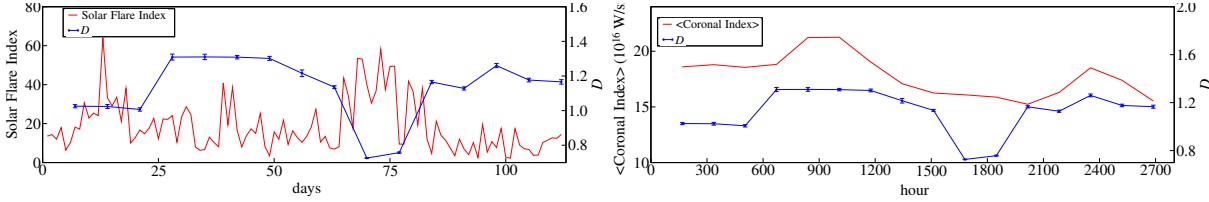

**Figure 9.** Box-counting dimension $D$ (with error bars) corresponding to the *Dst* index, along with the total solar flare (sum of northern and southern hemispheres indexes) (left panel) and coronal (right panel) indexes for the storms: 13 March 1989, with moving windows.

We observe that two different estimations of solar activity are correlated to some extent with $D$, thus suggesting a link between the solar activity and the fractal features of the Earth's magnetosphere. Certainly, one should probably not expect to find a single index to reveal this, as geomagnetic dynamics may be mostly but not exclusively determined by solar behavior, and several other correlated pairs have been proposed (Yurchyshyn et al., 2004), but it is interesting to notice the overall consistency of the results, at least when a correlation can be observed.

## 6  MHD Shell Model

Given the intrinsic difficulties in using direct numerical simulations to describe turbulent flows, specially for large Reynolds numbers, shell models have been used for years in order to reproduce the nonlinear dynamics of fluid systems in large dynamical ranges, but with less degrees of freedom (Obukhov, 1971; Gledzer, 1973; Yamada and Ohkitani, 1988). An MHD shell





model (Boffetta et al., 1999), in particular, is a dynamical system which aims to reproduce the main features of MHD turbulence. The model corresponds to a simplified version of the Navier-Stokes or MHD equations for turbulence, that conserves some of its invariants in the limit of no dissipation.

In this work, we use the MHD GOY shell model, which describes the dynamics of the energy cascade in MHD turbulence (Lepreti et al., 2004). The model is built up by dividing the wave-vector space ($k$-space) in $N$ discrete shells of radius $k_n = k_0 2^n$ ($n = 0, 1, \ldots, N$). Then, two complex dynamical variables $u_n(t)$ and $b_n(t)$ representing velocity and magnetic field increments on an eddy scale $l \sim k_n^{-1}$, are assigned to each shell.

The model consists of the following set of ordinary differential equations:

$$\frac{du_n}{dt} = -\nu k_n^2 u_n + ik_n \left( u_{n+1}u_{n+2} - b_{n+1}b_{n+2} \right) - ik_n \left\{ \frac{1}{4} \left( u_{n-1}u_{n+1} - b_{n-1}b_{n+1} \right) + \frac{1}{8} \left( u_{n-2}b_{n-1} - b_{n-2}u_{n-1} \right) \right\}^* + f_n \,, \tag{2}$$

$$\frac{db_n}{dt} = -\eta k_n^2 b_n + ik_n \frac{1}{6} \left( u_{n+1}b_{n+2} - b_{n+1}u_{n+2} \right) - ik_n \frac{1}{6} \left\{ \left( u_{n-1}b_{n+1} - b_{n-1}u_{n+1} \right) + \left( u_{n-2}b_{n-1} - b_{n-2}u_{n-1} \right) \right\}^* + g_n \,, \tag{3}$$

where $\nu$ and $\eta$ are, respectively, the kinematic viscosity and the resistivity; $f_n$ and $g_n$ are external forcing terms acting, respectively, on the velocity and magnetic fluctuations. The nonlinear terms have been obtained by imposing quadratic nonlinear coupling between neighbouring shells and the conservation of three MHD ideal invariants (Gloaguen et al., 1985; Lepreti et al., 2004).

The forcing terms are calculated according to the Langevin equation

$$\frac{d\tilde{f}_n}{dt} = -\frac{\tilde{f}_n}{\tau_0} + \tilde{\mu} \,, \tag{4}$$

where $\tilde{f}_n = f_n$ or $g_n$, $\tau_0$ is a characteristic time of the largest shell and $\tilde{\mu}$ is a Gaussian white noise of width $\sigma$.

The magnetic energy dissipation rate is defined as

$$\epsilon_b(t) = \eta \sum_{n=1}^{N} k_n^2 \left| b_n^2 \right| \,. \tag{5}$$

In our simulation, we set $\sigma = 0.01$, $\tau_0 = 0.25$, take $N = 19$ shells, and force the system on the largest shell ($f_1, g_1 \neq 0$). Similar parameters have been considered in previous studies using this model for modelling of solar flares statistics (Boffetta et al., 1999; Lepreti et al., 2004; Nigro et al., 2004).

We numerically integrate the shell model Eqs. (2)–(3) for various values of $\nu$ and $\eta$, and then we calculate the magnetic energy dissipation rate $\epsilon_b(t)$ (Eq. (5)).

Figure 10 shows a typical time behavior for $\epsilon_b(t)$.

Previous works have compared the statistics of bursts in turbulent systems with the statistics of dissipative events in the shell model (Boffetta et al., 1999; Lepreti et al., 2004; Carbone et al., 2002). There, peaks in the $\epsilon_b(t)$ time series have been associated to dissipative events in the magnetized plasma. Following the ideas in Ref. Domínguez et al. (2014), we focus only





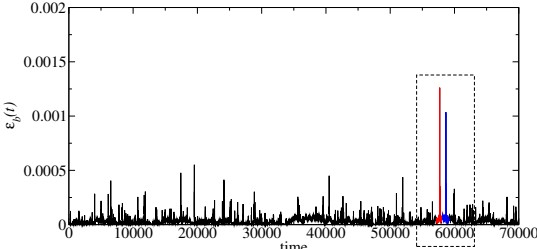

**Figure 10.** Time series of the energy dissipation rate $\epsilon_b(t)$, Eq. (5) for the shell model with $\nu = \eta = 10^{-4}$. The red and blue region inside the dashed box corresponds to an active state, as explained in Sec. 7.

on the largest peaks in the $\epsilon_b(t)$ time series, specifically, on dissipative events where the maximum value is larger than $\langle\epsilon_b\rangle + n\tilde{\sigma}$ where $\langle\epsilon_b\rangle$ is the average value of $\epsilon_b$ over all simulation time, $\tilde{\sigma}$ is the standard deviation of the $\epsilon_b$ time series in that window, and $n$ is a certain integer. In this paper we discuss only results for $n = 10$, but in Ref. Domínguez et al. (2017) $n = 5$ was also considered, in order to assess the robustness of the results. Our aim is to study the dependence of the conclusions on $\nu$ and $\eta$

in Eqs. (2) and (3).

## 7   Active and quiet states in the MHD shell model

We now apply the same techniques used to study the *Dst* index, as described in Secs. 3–8, to the $\epsilon_b(t)$ time series.

First, we need to define "active states" and "quiet states". However, unlike the *Dst* case (Domínguez et al., 2014), there is no clear criterion to establish the time scale of a typical dissipative event for our simulation data, and thus we proceed to inspect

the data. To this end, and in order to explore a wide range of parameters, we fix Pm $= \nu/\eta = 1$, and take values $\nu = \eta = 10^{-\mu}$ with $\mu = 1, 2, 3, \ldots, 12$. We then solve the shell model equations with a time step $dt = 10^{-4}$, for $7 \times 10^8$ iterations. This series of simulations suggest that $n = 10$ is enough to identify the largest peaks, filtering out most of the other events.

Regarding the width of an active states, Fig. 10 is, among the various simulations we performed, the only case where two clear dissipative events were both close and distinguishable from each other. Thus, this run was taken as a reference, and we

define an active state width such that both peaks in Fig. 10 can be regarded as two separate events. Since the separation between both peaks is 96 000 time steps, we will define an "active state" by identifying a peak, and then considering a window starting 48 000 time steps before, and ending 48 000 time steps after it. With this definition, in Fig. 10 we have two adjacent active states, each one associated with one of the peaks.

With these definitions of active and quiet states, we analyze the simulation results for $\nu = \eta = 10^{-\mu}$ with $\mu = 3$ with $n = 10$.

Three quiet states and two active states are identified. They are identified by integer numbers following the same strategy described in Sec. 3, states "2" and "4" corresponding to the active states.





Performing the procedure described in section 2, we calculate the scatter box-counting dimension for different values of $j$ for each active and quiet state. Results are shown in Figure 11. Errors bars in $D$ are given by the error of the least squares linear fit.

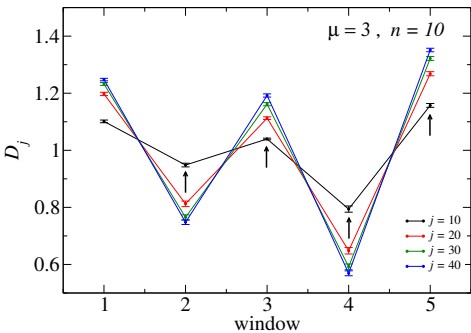

**Figure 11.** Box counting fractal dimension for $\epsilon_b(t)$ during quiet and active states for $n = 10$, with $\mu = 3$. Active states correspond to states labeled "2" and "4".

We note that in general, an active state has a smaller fractal dimension than the surrounding quiet states. This is observed for all values of $j$ considered, although quantitative differences occur. For instance, in Fig. 11 we note that when $j$ decreases the difference between quiet and active states is less clear.

5    Figure 11 also shows that the fractal dimension depends on the distance between consecutive data, represented by the parameter $j$, which may be seen as an indication of an underlying multifractal structure of the data (Kadanoff et al., 1995; Pisarenko et al., 1993). In order to further investigate the dependence on $j$, we plot the fractal dimension for each quiet and active state in the simulations as a function of the distance between data, $j$. Results are shown in Fig. 12.

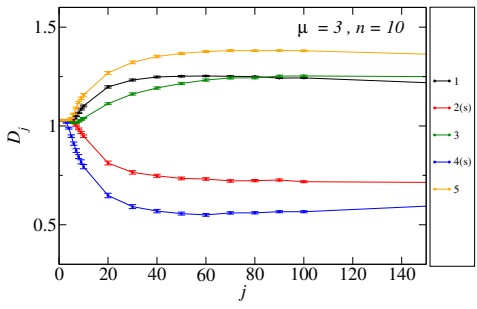

**Figure 12.** Box counting fractal dimension for quiet and active states, as a function of $j$, with $\mu = 3$. Numbers for each curve rotulate the states. We added an "(s)" in the legends, in order to highlight the active states.

As mentioned above, the scatter box fractal dimension when all data are taken ($j = 1$) is a straight line, yielding $D = 1$, consistent with Fig. 12. On the other hand, as $j$ increases, a smaller subset of simulation data is taken, and eventually, when $j$ is larger than the number of data, only one datum is taken, leading to $D = 0$. Both limits are found for all curves. For intermediate





values of $j$, a nontrivial dependence of the fractal dimension is observed, which also reflects the multifractal nature of the series as $D$ varies as we change the time scale given by $j$.

Figure 12 shows that active states have lower fractal dimensions than quiet states, consistently with Fig 11. Moreover, active states always have fractal dimensions less than 1, whereas it is always larger than 1 for quiet states. It also shows that all quiet (or equivalently all active) states are not characterized by a single fractal dimension, consistent with our previous findings.
Finally, Fig. 12 shows that the fractal dimension decreases during dissipative events, for a certain range of $j$. If $j > 1$, but not too large, the fractal dimension during active states is always smaller than during quiet times, which suggests that, for this range of moderate values of $j$, the box counting fractal dimension has statistical information on the activity of the time series.

In Ref. Domínguez et al. (2017) a more detailed analysis is carried out on the shell model results, exploring other simulation parameters ($\nu$, $\eta$, magnetic Prandtl number), other criteron for defining active states ($n = 5$), and a systematic study of the
correlations between the fractal dimension and the occurrence of dissipative events by means of the Student's $t$-test.

## 8   Magnetic clouds

As a way to illustrate how the ideas described so far could be used to characterize structures in space plasmas, we apply the method to study the time series for the magnetic field during magnetic clouds, (Burlaga et al., 1981) as found in ACE data (ACE Science Center). Magnetic clouds are transient structures ejected from the Sun, characterized by a large and smooth rotation
of the magnetic field. Typically, a magnetic cloud event can be identified from single spacecraft measurements by studying the evolution of the observed fields. During a given event, various stages can be identified: first, observation of solar wind prior to the cloud's arrival, then a sheath of compressed solar wind plasma immediately preceding a flux rope, where the magnetic field varies smoothly, and finally the background solar wind again. Note that slower-moving clouds traveling at speeds comparable to that of the ambient solar wind will not display prominent sheath regions.
Two events were selected: an event occuring on 12 July 2012 (MC1) and another on 11 July 2014 (MC2). Resolution for the magnetic field time series for this event is 16 seconds, covering a time span of 8 days for MC1 and 6 days for MC2, of which about 2 days correspond to the cloud event itself. It is found that the calculated fractal dimension evolves in a distinctive way as the various stages of the event as it passes by the spacecraft (namely surrounding solar wind, sheath, and flux rope). Given the high resolution of the data, it is possible to calculate the box-counting dimension for several delays, given by $j$, as was
shown in Figs. 11 and 12 for the shell model analysis. In Fig. 13 the fractal dimension is calculated for each magnetic cloud stage, and various values of the sampling $j$ are considered.

It can be noted that the fractal dimension, as calculated here, is indeed able to characterize magnetic cloud structures. The sheath state has a large dispersion of fractal dimension values as $j$ is varied, consistent with its more turbulent regime; on the other hand, the quieter and more organized flux rope state exhibits a very low variation with $j$, basically a single fractal
dimension at all time scales explored. As for the surrounding solar wind, it shows dispersion of $D_j$ which is between the dispersion of values in the sheath and the flux rope (Muñoz et al., 2016).



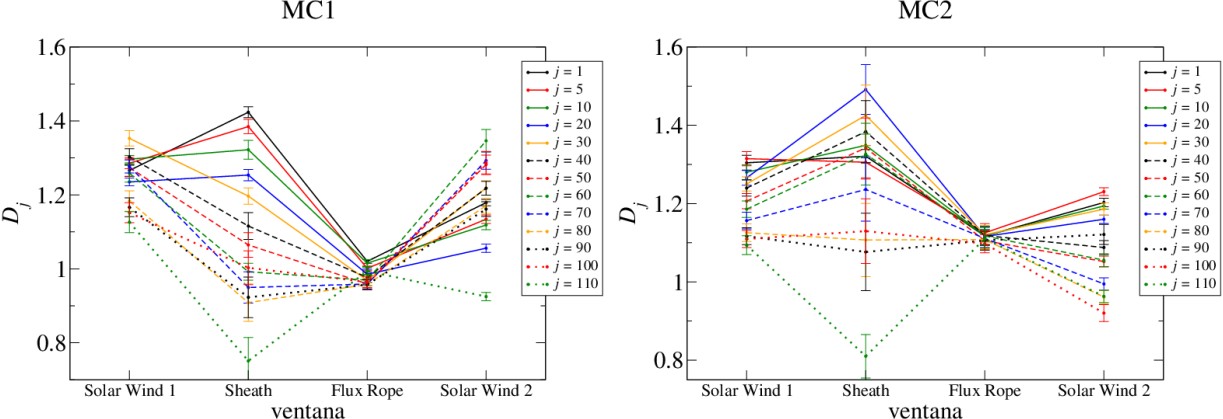

**Figure 13.** Box counting fractal dimension for two magnetic cloud events during the four stages of the time series: first the solar wind, then the sheath, then the flux rope, and finally the solar wind again. Several values for data sampling $j$ are used.

## 9 Conclusions

In this manuscript, we have reviewed recent results obtained by us, regarding the evolution of complexity in magnetized plasmas, as described by geomagnetic data, simulation results for MHD turbulence, and spacecraft data in the solar wind.

This has been done by calculating a box-counting fractal dimension for time series of magnetic field data for the *Dst* geomagnetic index (Domínguez et al., 2014), the GOY shell model (Domínguez et al., 2017), and ACE data for two magnetic cloud events (Muñoz et al., 2016).

    Some robust behaviors are identified. In general, it is found that the fractal dimension $D$ decreases during dissipative events. In the case of the *Dst* time series this was verified for three different types of time windows: fixed width and stationary (Sec. 3),

variable width (Sec. 4), and moving windows (Sec. 8). And it was also found across several time scales, namely individual storms, full years, and the complete 23rd solar cycle, as detailed in Ref. Domínguez et al. (2014).

    A similar behavior is found for the MHD shell model (Secs. 6 and 7). Thanks to the larger resolution of the simulation data as compared with the *Dst* data, several values of the time delay for data sampling could be made, showing that the results found in Ref. Domínguez et al. (2014) are nontrivial, in the sense that not all samplings yield similar results. Only intermediate, not

too large, values of the time delay (as represented by the value of $j$ in Sec. 7 are able to clearly distinguish between active and quiet states. But, within the useful range of values for $j$, the fractal dimension of the active states is consistently smaller than the dimension of quiet states, and is always lower than 1, whereas the active states always have a dimension larger than 1. The dependence on $j$ of the fractal dimension is interesting in itself, as it suggests that data have a multifractal structure, which is consistent with suggestions and finding by other authors for space plasmas (Chapman et al., 2008, 1998; Valdivia et al., 2005).





Also, a more systematic test for the correlation between burst events in the shell model and the decrease in fractal dimension was performed, by means of the Student's $t$-test, as well as a more detailed exploration of the parameter space for the simulation. These results can be found in Ref. (Domínguez et al., 2017).

As an application of these ideas, we take two magnetic cloud events in the solar wind, and use the techniques described here to study the corresponding magnetic field time series. Our results, although preliminary, suggest that this method can characterize the various stages of the magnetic cloud structure.

     Given the rich and complex dynamics governing the evolution of magnetized plasmas, we would not expect that a single index would be able to capture all their relevant information. However, the findings summarized here suggest that some relevant

correlations can be observed, and that the dimension used here, although simple, may give some insight on the evolution of complexity of plasmas in the Sun-Earth system and MHD turbulent states.

*Competing interests.* The authors declare that they have no conflict of interest.

*Acknowledgements.* This project has been financially supported by FONDECYT under contracts Nos. 1110135, 1110729, and 1130273 (J.A.V.); Nos. 1080658, 1121144, and 1161711 (V.M.); and No. 3160305 (M.D.). M.D. also thanks a doctoral fellowship from CONICYT,

and a Becas-Chile doctoral stay, contract No. 7513047. We are also thankful for financial support by CEDENNA (J.A.V.), and the US AFOSR Grant FA9550-16-1-0384 (J.A.V. and V.M.).





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
