# Peer review of "Evolution of fractality in magnetized plasmas"

_Nonlinear Processes in Geophysics, 2017_

## Referee Comment (RC1) · Anonymous Referee #1 · 1 Sep 2017

The central idea of this review is very interesting. However, there is a major problem with the present manuscript. The similarity report showed a very high similarity index (26%) resulting from a single reference (Dominguez et al., 2014)! Unless this issue is resolved I would not be able to perform a thorough review of this manuscript. I would be happy to provide specific comments when a suitably revised version of this paper will be resubmitted.

---

## Referee Comment (RC2) · Anonymous Referee #2 · 2 Sep 2017

The manuscript features a nice study of fractality in timeseries, describing a way to expose intermittent behavior by means of the scatter diagram of Figure 1. Then the manuscript shows conclusively that in cases of intermittency the fractal dimension decreases. This is the case for Dst and solar-flare index timeseries (Sections 3 – 5), one-dimensional MHD turbulence shell models (Sections 6, 7) and two cases of magnetic clouds (Section 8).

Despite the intuitive, valid fractal analysis, I feel that the manuscript does not have many new elements to showcase. The reference to turbulence in efforts to physically connect solar, interplanetary, and magnetospheric timeseries is biased in its framework. The manuscript effectively shows the effect of intermittency in the fractal dimension of timeseries, regardless of turbulence. Intermittency is a term that is broader than turbulence: turbulent timeseries may be intermittent, but not all intermittent timeseries stem from turbulent systems.

The scatter diagram of Figure 1 creates some "dust-like" fractals in case of intermittency (in this case, storm-time dips in Dst). Dust-like structures typically give rise to a fractal dimension smaller than Dmax – 1, where Dmax is the embedding (i.e., Euclidean) dimension of the studied space. In Figure 1, Dmax=2, hence the dust-like structures in the lower-left part of the image show a fractal dimension D < 1 (see, e.g., Schroeder, M,: Fractal, Chaos, Power Laws. Minutes from an Infinite Paradise, Freeman, New York, NY). If no significant intermittency is present, one is left with the upper right part of Figure 1 that typically gives 1 < D < 2.

Interpreting intermittency in general as turbulence and drawing physical conclusions from it is the main drawback of the manuscript. This leads to insufficiently justified conclusions such as the correlations between D from Dst timeseries and the solar flare / coronal indices over tens of days (Figure 9). Indeed, there is connection if an eruptive flare (flare + coronal mass ejection) leads to a magnetospheric storm within 1 – 3 days. However, the correlation seen in Figure 9 is not due to physics but due to the fact that any two intermittent timeseries with intermittent excursions roughly matching in time will show similar correlations. I am afraid this is a common fallacy, appearing in several interdisciplinary studies of timeseries giving, not surprisingly, incidental correlations.

Another unjustified conclusion is the one drawn from Figures 7, 8, namely that "results suggest that the box-counting dimension consistently decreases when the storm approaches" (p.8; top). However, the decrease is not due to the storm but due to the pre-storm disturbances (hours > 1400 and up to the storm's onset). These disturbances are not necessarily related to the storm. Similar disturbances appear at times < 500 hours in the absence of a storm. Not surprisingly, D in this interval is very similar to the pre-storm D that is indeed decreasing. Again, it is the (most likely incidental, as it starts ∼300 hours prior to the storm) minor intermittency in the timeseries that causes the decrease in both cases, regardless of the storm. Finding a unique pre-storm signature is the challenge here and the manuscript does not seem to contribute significantly to this cause.

The above issues render the penultimate conclusion of the manuscript (p.14) also biased. I see no point in re-doing the analysis unless more physical and statistical arguments for the apparent correlations are used alongside the analysis of the fractal dimension.

---

## Referee Comment (RC3) · Anonymous Referee #3 · 4 Sep 2017

In my view, the submitted manuscript is interesting and possibly worth publishing in Nonlinear Processes in Geophysics, but after some revisions, and with more specific title and somewhat weaker conclusions. The similarity index of 28% could be acceptable for a review provided that all credits are given, even if the authors of the previous published papers are also on the authors list of the review. But 26% (including Figures 1 and 2) are simply copied from Dominguez et al. (2014).

Obviously, as mentioned in the introduction fractal dimensions have already often been calculated for space and laboratory magnetized plasmas in nature, including the magnetosphere (e.g., J. Geophys. Res. 96, 16031, 1991) and the solar wind (e.g., J. Geophys. Res. 114, A03108, 2009; Astrophys. J. Lett., 793:L30, 2014). But the subject of the submitted review is rather limited to very selected examples of space plasmas, basically only to geomagnetic activity (besides preliminary results applied to magnetic clouds and additional discussion in the context of the turbulence shell model)

[Figure]

and therefore the title of the review should possibly be much more specific.

By the way, the phenomenological MHD shell model describes the energy cascade in turbulence that sometimes exhibits fractal characteristics, but geomagnetic storms have quite different more intermittent characters, sometimes related to multifractality. It would be nice to provide convincing physical arguments justifying application of this model to dynamics of geomagnetic activity.

Please find also my specific comments:

page 3, lines 16-18: Admittedly, there is no commonly accepted definition of a fractal (for example, according to B. B. Mandelbrot, 1977: 'a fractal is by definition as set for which the Hausdorff Besicovitch dimension strictly exceeds the topological dimension'). But certainly, 'noninteger numbers measuring the complexity' is rather unclear (maybe roughness, irregularity) and certainly not general (e.g., for the trail fractal Brownian motion its fractal dimension is integer, equal to 2, but greater than 1, the topological dimension).

Section 2: The methods of nonlinear time series are well-known, see e.g. the textbook of H. Kantz and T. Schreiber published by Cambridge University Press in 1997. Besides the box-counting (zero-order, capacity) dimension one can also define the (higher-order) generalized dimensions (related to a multifractal spectrum), which are (e.g., the correlation dimension) much more suitable for nonlinear dynamical systems as is in the case of the magnetosphere. Therefore, I would like to ask why the authors use only the box-counting method, which is certainly not very reliable?

Further, for estimation of any fractal dimension one would require at least approximate stationarity. Hence, my main question is how do the authors cope with non-stationarity of the data under their study, especially during storms. I think that in the magnetospheric studies it would be more difficult task than in the case of the solar wind plasma. Maybe also some filtering is needed before estimating the actual dimension of the fractal structure (see, e.g.: Phys. Rev. E 47, 2401, 1993; Physica D 122, 254, 1998).

Results and Conclusions:

Relation of the fractal dimensions to storms should be better justified. Namely, a decrease of the fractal dimension based on Dst index presented in Figures 8 and 9 during storms may simply artificially result from lack of stationarity. Anyway, a more comprehensive nonlinear time series analysis is needed before drawing any robust conclusion (e.g., page 13, line 8ff).

Please also note the supplement to this comment:
https://www.nonlin-processes-geophys-discuss.net/npg-2017-47/npg-2017-47-RC3-supplement.pdf

---

## Author Comment (AC1) · 28 Nov 2017

**Response to the First Referee.**

First, we would like to thank the Referee for her/his comment.

The referee has made a point that the similarity report is too high even for a review article. We have addressed this issue, rewriting several portions of the manuscript, refering to the relevant references where necessary, and dropping some paragraphs not necessary for this review's discussion. We hope that the manuscript is now in a more satisfactory state.

---

## Author Comment (AC2) · 28 Nov 2017

**Response to the Second Referee.**

First, we would like to thank the Referee for her/his comments, all of which we have attempted to address. We think that the paper has been improved by them. Now we detail our response to each comment.

1. **Despite the intuitive, valid fractal analysis, I feel that the manuscript does not have many new elements to showcase. The reference to turbulence in efforts to physically connect solar, interplanetary, and magnetospheric timeseries is biased in its framework. The manuscript effectively shows the effect of intermittency in the fractal dimension of timeseries, regardless of turbulence. Intermittency is a term that is broader than turbulence: turbulent timeseries may be intermittent, but not all intermittent timeseries stem from turbulent systems.**

   **The scatter diagram of Figure 1 creates some "dust-like" fractals in case of intermittency (in this case, storm-time dips in Dst). Dust-like structures typically give rise to a fractal dimension smaller than Dmax=1, where Dmax is the embedding (i.e., Euclidean) dimension of the studied space. In Figure 1, Dmax=2, hence the dust-like structures in the lower-left part of the image show a fractal dimension $D < 1$ (see, e.g. Schroeder, M,: Fractal, Chaos, Power Laws. Minutes from an Infinite Paradise, Freeman, New York, NY). If no significant intermittency is present, one is left with the upper right part of Figure 1 that typically gives $1 < D < 2$.**

   **Interpreting intermittency in general as turbulence and drawing physical conclusions from it is the main drawback of the manuscript.**

   It was not our intention to force a connection between all three systems studied through the concept of turbulence. As stated in the manuscript, the GOY shell model has been shown to exhibit dissipative events whose distribution follows the same power-law statistics as observed in turbulent magnetized plasmas (Boffetta et al., 1999; Lepreti et al., 2004; Carbone et al., 2002), and our main goal was to test whether such

bursty behavior exhibited fractal features similar to those found in the *Dst* analysis.

Although various works suggest the presence of turbulence in the Earth's magnetosphere, the question of the validity of the GOY shell model to describe such phenomena is far beyond the scope of our paper. As the referee correctly points out, it is the intermittency of the time series which is the relevant feature, and in fact that is what justifies the use of certain values of $\nu$ and $\eta$ for the simulations, since, in general, intermittency levels similar to the *Dst* timeseries are not observed for arbitrary values of these parameters.

We have attempted to clarify this issue in various parts of the text. For instance, in the second paragraph of Sec. 4.

The new text reads:

*We first notice that, in general, setting parameters $\nu$ and $\eta$ with arbitrary values yields $\epsilon_b(t)$ series which do not have the necessary intermittency level to resemble the Dst time series. Compare, for instance, the different panels in Fig. 16 in Domínguez et al. (2017), which shows that $Pm = 0.2$ leads to a very noisy output, unlike simulations with $Pm = 1.0$ or 2.0, where individual, large peaks can be easily identified from the background. In fact, previous studies have shown that the statistics of bursts follows a power law for $Pm = 1$ (Boffetta et al., 1999; Lepreti et al., 2004; Carbone et al., 2002), and for this reason we start by taking $Pm = \nu/\eta = 1$.*

Also, in the final paragraph of the same section.

The new text reads:

*Results suggest that the intermittency level of the output time series is relevant, which has led us to perform the analyses for the shell model within a certain range of values of the Prandtl number, as well as of the viscosity and resistivity.*

2. **This leads to insufficiently justified conclusions such as the correlations between D from Dst timeseries and the solar flare**

/ coronal indices over tens of days (Figure 9). Indeed, there is connection if an eruptive flare (flare + coronal mass ejection) leads to a magnetospheric storm within 1  3 days. However, the correlation seen in Figure 9 is not due to physics but due to the fact that any two intermittent timeseries with intermittent excursions roughly matching in time will show similar correlations. I am afraid this is a common fallacy, appearing in several interdisciplinary studies of timeseries giving, not surprisingly, incidental correlations.

Thanks for pointing out this issue. The figure that the referee mentions (Fig. 9 in the previous version of the manuscript), was part of an exploration of possible correlations between fractal dimensions and various indices, using solar and geomagnetic timeseries, which was made in Domínguez et al. (2014). As the referee says, a better statistical and physical analysis is needed to state whether these correlations hold or not. Besides, in the context of the present manuscript, it is not a relevant discussion, since we focus on the series themselves, not on their correlations with others. We have thus dropped Fig. 9 in this version of the manuscript.

3. **Another unjustified conclusion is the one drawn from Figures 7, 8, namely that "results suggest that the box-counting dimension consistently decreases when the storm approaches" (p.8; top). However, the decrease is not due to the storm but due to the pre-storm disturbances (hours > 1400 and up to the storm's onset). These disturbances are not necessarily related to the storm. Similar disturbances appear at times < 500 hours in the absence of a storm. Not surprisingly, D in this interval is very similar to the pre-storm D that is indeed decreasing. Again, it is the (most likely incidental, as it starts ~300 hours prior to the storm) minor intermittency in the timeseries that causes the decrease in both cases, regardless of the storm. Finding a unique pre-storm signature is the challenge here and the manuscript does not seem to contribute significantly to this cause.**

We agree that conclusions need to be toned down, and that the present analysis cannot suggest that the decrease observed before the storm is related to the storm itself. However, our aim in this manuscript is focused rather on the dissipative events themselves and the fractal dimension, not on the finding of precursors for geomagnetic activity, an issue which requires further, detailed analysis.

Thus, we have changed the wording in the sentence mentioned (now at the bottom of page 5).

The new text reads:

*As shown in Domínguez et al. (2014), the box-counting dimension of the Dst index decreases as the storm approaches for all cases studied. Moreover, this decrease occurs before the window includes the geomagnetic storm, as marked by the vertical lines in Fig. 5. Whether this is relevant for forecasting geomagnetic storm needs further study, as it may simply be due to an increase of the intermittency in the time series, unrelated to the upcoming dissipative event.*

4. **The above issues render the penultimate conclusion of the manuscript (p.14) also biased. I see no point in re-doing the analysis unless more physical and statistical arguments for the apparent correlations are used alongside the analysis of the fractal dimension.**

We have indeed performed more systematic analysis than the ones mentioned in this manuscript, but were left in the cited references and not included in the current text.

Cross correlation analyses between the *Dst* timeseries and its fractal dimension were performed. This is not a direct calculation, as both time series have different resolutions, and thus interpolation of the fractal dimension time series is needed to match the resolution of the geomagnetic index. This analysis was made for individual storms and full year data, and is included in Domínguez et al. (2014). This is mentioned in the final paragraph of Sec. 3.

On the other hand, *p*-value analyses were systematically done for the

shell model simulations, for a wide range of values of $\nu$ and $\eta$, considering Pm $= 1$ and Pm $\neq 1$. This allowed us to find a range of values of the simulation parameters where the correlation between $\epsilon_b(t)$ and its fractal dimension is statistically significant.

We have mentioned this issue in the first paragraph of page 8, relating it to the problem of intermittency.

The new text reads:

*We first notice that, in general, setting parameters $\nu$ and $\eta$ with arbitrary values yields $\epsilon_b(t)$ series which do not have the necessary intermittency level to resemble the Dst time series. Compare, for instance, the different panels in Fig. 16 in Domínguez et al. (2017), which shows that $Pm = 0.2$ leads to a very noisy output, unlike simulations with $Pm = 1.0$ or 2.0, where individual, large peaks can be easily identified from the background. In fact, previous studies have shown that the statistics of bursts follows a power law for $Pm = 1$ (Boffetta et al., 1999; Lepreti et al., 2004; Carbone et al., 2002), and for this reason we start by taking $Pm = \nu/\eta = 1$.*

And again in the last paragraph of Sec. 4.

The new text reads:

*In Domínguez et al. (2017) a more detailed analysis is carried out on the shell model results, exploring other simulation parameters ($\nu$, $\eta$, magnetic Prandtl number), other criteron for defining active states ($n = 5$), and a systematic study of the correlations between the fractal dimension and the occurrence of dissipative events by means of the Student's t-test. Results suggest that the intermittency level of the output time series is relevant, which has led us to perform the analyses for the shell model within a certain range of values of the Prandtl number, as well as of the viscosity and resistivity.*

---

## Author Comment (AC3) · 28 Nov 2017

**Response to the Third Referee.**

First, we would like to thank the Referee for her/his comments, all of which we have attempted to address. We think that the paper has been improved by them. Now we detail our response to each comment.

1. **The similarity index of 28% could be acceptable for a review provided that all credits are given, even if the authors of the previous published papers are also on the authors list of the review. But 26% (including Figures 1 and 2) are simply copied from Dominguez et al. (2014).**

   We have made several modifications in various parts of the manuscript in order to deal with this issue, including dropping parts of the text that were not relevant for the line of the discussion intended in this paper.

2. **Obviously, as mentioned in the introduction fractal dimensions have already often been calculated for space and laboratory magnetized plasmas in nature, including the magnetosphere (e.g., J. Geophys. Res. 96, 16031, 1991) and the solar wind (e.g., J. Geophys. Res. 114, A03108, 2009; Astrophys. J. Lett., 793:L30, 2014). But the subject of the submitted review is rather limited to very selected examples of space plasmas, basically only to geomagnetic activity (besides preliminary results applied to magnetic clouds and additional discussion in the context of the turbulence shell model) and therefore the title of the review should possibly be much more specific.**

   We have changed the title to "Evolution of fractality in space plasmas of interest to geomagnetic activity", in order to be more specific and consistent with the content of the manuscript.

3. **By the way, the phenomenological MHD shell model describes the energy cascade in turbulence that sometimes exhibits fractal characteristics, but geomagnetic storms have quite different more intermittent characters, sometimes related to multifractality. It would be nice to provide convincing physical**

**arguments justifying application of this model to dynamics of geomagnetic activity.**

Maybe we should stress that we are not attempting to use the MHD shell model to account for *Dst* dynamics. Our interest in the connection between two model arises from the possibility of having similar intermittent behaviors, as the shell model can also yield simulations which do not exhibit intermittency levels which resemble the *Dst* time series.

We have added a text in the first paragraph of page 8, related to this issue.

The new text reads:

*We first notice that, in general, setting parameters $\nu$ and $\eta$ with arbitrary values yields $\epsilon_b(t)$ series which do not have the necessary intermittency level to resemble the Dst time series. Compare, for instance, the different panels in Fig. 16 in Domínguez et al. (2017), which shows that $Pm = 0.2$ leads to a very noisy output, unlike simulations with $Pm = 1.0$ or 2.0, where individual, large peaks can be easily identified from the background. In fact, previous studies have shown that the statistics of bursts follows a power law for $Pm = 1$ (Boffetta et al., 1999; Lepreti et al., 2004; Carbone et al., 2002), and for this reason we start by taking $Pm = \nu/\eta = 1$.*

We have also been careful in the use of words, refering to dissipative events in the shell model as "active" states, whereas in the *Dst* time series they correspond to "storm" states, with definite physical meaning.

The possible connection between geomagnetic activity and the GOY shell model has been suggested in Lepreti et al. (2004), but testing this goes beyond the simple fractal analysis we propose in this manuscript.

4. **page 3, lines 16-18: Admittedly, there is no commonly accepted definition of a fractal (for example, according to B. B. Mandelbrot, 1977: "a fractal is by definition as set for which the Hausdorff Besicovitch dimension strictly exceeds the topo-**

logical dimension"). But certainly, "noninteger numbers measuring the complexity" is rather unclear (maybe roughness, irregularity) and certainly not general (e.g., for the trail fractal Brownian motion its fractal dimension is integer, equal to 2, but greater than 1, the topological dimension).

We agree with the Referee in that one has to be careful with definitions. However, we should notice that the cited sentence in our paper refers to the problem of defining fractal *dimensions*, rather than fractal objects. So, for a given fractal object, there are several ways to define its dimension, and this is what we intended to stress. We have modified the sentence to be more clear.

The new text reads:

*In general it can be said that they are numbers, which can be noninteger, measuring the complexity of a data set.*

5. **Section 2: The methods of nonlinear time series are well-known, see e.g. the textbook of H. Kantz and T. Schreiber published by Cambridge University Press in 1997. Besides the box-counting (zero-order, capacity) dimension one can also define the (higherorder) generalized dimensions (related to a multifractal spectrum), which are (e.g., the correlation dimension) much more suitable for nonlinear dynamical systems as is in the case of the magnetosphere. Therefore, I would like to ask why the authors use only the box-counting method, which is certainly not very reliable?**

Our aim was specifically to investigate whether a single fractal dimension may yield useful information on the systems studied, and in what sense. Certainly, given the complexity of the system, there is no guarantee that this is possible at all, but we have found some positive results as described in the manuscript, which we think are interesting. Other choices for that single fractal dimension could have been made. However, rather than changing the type of dimension used, we think it is

more interesting to perform a multifractal analysis, in accordance with the nature of the systems studied, and this is currently in process.

This is mentioned in the final paragraph of Sec. 7.

The new text reads:

*Given the rich and complex dynamics governing the evolution of magnetized plasmas, we would not expect that a single index would be able to capture all their relevant information. In fact, multifractal analysis should be made in order to represent the dynamics of the systems studied more accurately, and such an analysis is currently being prepared for future publication.*

6. **Further, for estimation of any fractal dimension one would require at least approximate stationarity. Hence, my main question is how do the authors cope with non-stationarity of the data under their study, especially during storms. I think that in the magnetospheric studies it would be more difficult task than in the case of the solar wind plasma. Maybe also some filtering is needed before estimating the actual dimension of the fractal structure (see, e.g.: Phys. Rev. E 47, 2401, 1993; Physica D 122, 254, 1998).**

It is not clear that, for the kind of analysis we are interested, stationarity is a requisite to get meaningful results. For instance, the magnetic cloud analysis clearly involves a process where various degress of stationarity are found. Without looking at the fractal dimension, one could argue that the flux rope stage satisfies the stationarity criterion, the sheath does not, and the solar wind stages could also be approximately stationary. And yet, calculation of the fractal dimension on each state, regardless of its level of stationarity, yields useful results, being able to distinguish the various stages.

This is because the fractal dimension that we calculate is related to the intermittency level of the time series, which is also why storms leave a signature in the dimension, a signature which could be lost with filtering, as suggested by Fig. 8 in our paper. The issue of the need

for stationarity in the *Dst* or shell model time series should be studied more systematically in order to give a definitive answer.

7. **Results and Conclusions: Relation of the fractal dimensions to storms should be better justified. Namely, a decrease of the fractal dimension based on Dst index presented in Figures 8 and 9 during storms may simply artificially result from lack of stationarity. Anyway, a more comprehensive nonlinear time series analysis is needed before drawing any robust conclusion (e.g., page 13, line 8ff).**

We have attemted to tone down the conclusion in this respect. Figure 9 of the previous manuscript has been dropped from the current version of the manuscript, since it was not relevant to the main discussion. Regarding Fig. 8 in the previous version (Fig. 5 in the current one), it is true that the decrease in the fractal dimension previous to the storm could be due to pre-storm intermittency unrelated to the upcoming storm. However, our aim in this manuscript is focused rather on the dissipative events themselves and the fractal dimension, not on the finding of precursors for geomagnetic activity, an issue which requires further, detailed analysis.

Thus, we have changed the wording in the sentence mentioned (now at the bottom of page 5).

The new text reads:

*As shown in Domínguez et al. (2014), the box-counting dimension of the Dst index decreases as the storm approaches for all cases studied. Moreover, this decrease occurs before the window includes the geomagnetic storm, as marked by the vertical lines in Fig. 5. Whether this is relevant for forecasting geomagnetic storm needs further study, as it may simply be due to an increase of the intermittency in the time series, unrelated to the upcoming dissipative event.*

Also, the "Some robust behaviors are identified" sentence in the conclusions has been dropped, in order to moderate the conclusions.

---

## Referee Report (RR1)

Dear Editor,

I have reviewed the revised version of the manuscript (MS) "Evolution of fractality in space plasmas of interest to geomagnetic activity" by Muñoz et al. submitted for possible publication in *Nonlinear Processes in Geophysics (NPG)* and found that it may be acceptable for publication in *NPG* following intermediate revisions. This MS presents the results from two recent papers published by the same authors (Domínguez et al., 2014, 2017) on fractal dimension analysis and MHD model simulation of Dst index, respectively. This is done in the first 4 sections and 8 figures of the present MS, while in the penultimate section before Conclusions and in the 9th figure of the MS they apply the former ideas on solar wind data around magnetic clouds. The latter are the only new results of the MS and it is a pity that the authors did not devote some more space to the analysis and discussion of the results from the solar wind data. Additionally, they keep referring to an abstract published in conference proceedings by Muñoz et al. (2016) to downgrade the originality of their findings. **So, I believe that this last reference should be omitted from the MS and the authors should spend a few more words on the magnetic clouds' analysis and results in order to justify the acceptance of their work in *NPG*.**

*Comments*

1. The MS largely summarizes the results from 2 recent papers by the authors using text and 8 figures from these papers. There is only 1 new section and 1 new figure for the magnetic cloud case in the whole MS. Moreover, the originality of the new results is limited by referring to a conference abstract by Muñoz et al. (2016). I would suggest to the authors to omit the reference by Muñoz et al. (2016), both in the text and References, and devote more text and figures for the magnetic cloud case (Section 5). For instance, they could at least explain what exactly solar wind data they analyze (are they IMF data?) and provide a new figure showing the corresponding solar wind time series. They could also discuss a bit more the details of the MC analysis and elaborate on the importance of their MC findings.

2. Section 3:
   The author should clearly state here that by setting a threshold of -200 nT for the storm events (Figure 2) they focus on intense magnetic storms.

3. Section 3:
   "Similar plots for 5 years of high geomagnetic activity were obtained (Domínguez et al., 2014). In general, storm states are found to have smaller fractal dimension than quiet states immediately before and after them, although there does not seem to be a clear correlation on the value of Dst itself (Domínguez et al., 2014). Thus, our statement on the decrease of the fractal dimension is an argument on its variation, rather than on its actual value."
   This result agrees with the decrease in Tsallis entropy of the Dst index time series around intense magnetic storms found by Balasis et al. (2008).

*Remarks*

Abstract: please define the abbreviation "MHD"

Introduction:

Page 1, line 10: "leads" ---> "leading"

Page 1, lines 11-12: "of forecasting of space weather" ---> "of forecasting space weather"

Page 1, line 17: "broder" ---> "broader"

Page 2, line 22: please define the abbreviation "MHD"

Page 2, line 33: "magnetohydrodynamic (MHD)" ---> "MHD"

Section 2:

Page 3, line 16: please provide a link for the Dst index in the parentheses

Page 4, line 10: "versus de" ---> "versus the"

Section 4:

Page 6, line 12: please define the abbreviation "GOY"

Equations (2) and (3): please define the meaning of the "*" symbols

Page 7, lines 16-17: "Following the ideas in Domínguez et al. (2014), we focus only on the largest peaks in the ... time series". Why? What are these ideas? Please provide more explanations here

Page 9, line 12: "criteron" ---> "criterion"

Figure 9: what is the meaning of "ventana" in x-axis? please provide a proper label

Section 5:

Page 9, lines 18-19: please provide a link for ACE data in the parentheses

Conclusions:

Page 11, line 5: "variable width (Sec. ??)", please provide the correct number here

*References*

Balasis, G., I. A. Daglis, C. Papadimitriou, M. Kalimeri, A. Anastasiadis, and K. Eftaxias (2008), Dynamical complexity in Dst time series using non-extensive Tsallis entropy, Geophys. Res. Lett., 35, L14102, doi:10.1029/2008GL034743.

Sincerely,

---

## Author Response (AR2)

**Response to the First Referee.**

We thank again the comments by the First Referee. We have attempted to address them all, and detail our responses in the following.

1. **The MS largely summarizes the results from 2 recent papers by the authors using text and 8 figures from these papers. There is only 1 new section and 1 new figure for the magnetic cloud case in the whole MS. Moreover, the originality of the new results is limited by referring to a conference abstract by Muoz et al. (2016). I would suggest to the authors to omit the reference by Muoz et al. (2016), both in the text and References, and devote more text and figures for the magnetic cloud case (Section 5). For instance, they could at least explain what exactly solar wind data they analyze (are they IMF data?) and provide a new figure showing the corresponding solar wind time series. They could also discuss a bit more the details of the MC analysis and elaborate on the importance of their MC findings.**

   We have omitted the reference, and have added more explanations in the magnetic cloud section. For instance, we have clarified the location of the ACE spacecraft.

   The new text reads:

   *As a way to illustrate how the ideas described so far could be used to characterize structures in space plasmas, we apply the method to study the time series for the magnetic field during magnetic clouds (?), as found in ACE interplanetary magnetic field data (ACE Science Center,* `http://www.srl.caltech.edu/ACE/ASC/index.html`*), and measured in the proximity of the L1 Lagrangian point.*

   We have added the new Fig. 9 to show the magnetic field time series analyzed.

   And we have added two paragraphs at the end of the section commenting on the results.

The new text reads:

*The results above suggest that, from the point of view of the time series, the level of multifractality is large in the sheath, consistent with its more turbulent nature, intermediate in the solar wind, and that the flux rope magnetic field is essentialy monofractal, consistent with the organized, smoother structure of the magnetic field expected in this region. We plan to carry out other multifractal analyses to complement these findings in a future publication.*

*Also, these results suggest that the fractal approach discussed in this paper may be useful to characterize the various stages of magnetic clouds, and in particular to setup a system automatically identify similar magnetic structures in spacecraft data.*

2. **The author should clearly state here that by setting a threshold of -200 nT for the storm events (Figure 2) they focus on intense magnetic storms.**

We have added text to specify this.

The new text reads:

*Following ?, we are interested in testing the usefulness of the method by first stuyding strong, clear events. Thus, we identify "storm states" and "quiet states" by locating Dst peaks, such that $Dst < -220 \ nT$, which corresponds to intense magnetic storms.*

3. **"Similar plots for 5 years of high geomagnetic activity were obtained (Domínguez et al., 2014). In general, storm states are found to have smaller fractal dimension than quiet states immediately before and after them, although there does not seem to be a clear correlation on the value of Dst itself (Domínguez et al., 2014). Thus, our statement on the decrease of the fractal dimension is an argument on its variation, rather than on its actual value." This result agrees with the decrease in Tsallis entropy of the Dst index time series around intense**

**magnetic storms found by Balasis et al. (2008).**

We thank the Referee for the additional reference, and we have include text to acknowledge it.

The new text reads:

*These results are consistent with **?**, where it is shown that the Tsallis entropy of the Dst time series decreases during intense magnetic storms (Dst < −150 nT in their case).*

We also thank the Referee for all the other remarks. We have corrected all typos and provided definitions for the acronyms before their first use. Regarding some specific remarks:

1. **Page 3, line 16: please provide a link for the Dst index in the parentheses**

   **Page 9, lines 18-19: please provide a link for ACE data in the parentheses**

   We have provided the links explicity in the main text for both references.

2. **Page 6, line 12: please define the abbreviation "GOY"**

   The new text reads:

   *In this work, we use the MHD shell model proposed by Gledzer, Okhitani and Yamada (GOY shell model)*

3. **Equations (2) and (3): please define the meaning of the "*" symbols**

The new text reads:

*The symbol \* represents a complex conjugate quantity.*

4. **Page 7, lines 16-17: "Following the ideas in Domínguez et al. (2014), we focus onl on the largest peaks in the ... time series". Why? What are these ideas? Please provid more explanations here**

We have changed the text to refer to the contents of the section on geomagnetic storms and make it self-consistent.

The new text reads:

*Following the ideas in Sec. 3 regarding the size of events considered, we focus only on the largest peaks in the $\epsilon_b(t)$ time series*

5. **Figure 9: what is the meaning of "ventana" in x-axis? please provide a proper label**

We have changed the picture. Now the $x$-axis says "window". We apologize for the mistake.

6. **Page 11, line 5: "variable width (Sec. ??)", please provide the correct number here**

The text refers to three types of time windows, and in the current version needed only to refer to Sec. 3. The text was changed accordingly.

The new text reads:

[revised manuscript text omitted]